# Optimization Methods of Tungsten Oxide-Based Nanostructures as Electrocatalysts for Water Splitting

**DOI:** 10.3390/nano13111727

**Published:** 2023-05-25

**Authors:** Yange Wang, Rongming Wang, Sibin Duan

**Affiliations:** Beijing Advanced Innovation Center for Materials Genome Engineering, Beijing Key Laboratory for Magneto-Photoelectrical Composite and Interface Science, State Key Laboratory for Advanced Metals and Materials, School of Mathematics and Physics, University of Science and Technology Beijing, Beijing 100083, China

**Keywords:** tungsten oxide, electrocatalysis, structure–property relationship, water splitting

## Abstract

Electrocatalytic water splitting, as a sustainable, pollution-free and convenient method of hydrogen production, has attracted the attention of researchers. However, due to the high reaction barrier and slow four-electron transfer process, it is necessary to develop and design efficient electrocatalysts to promote electron transfer and improve reaction kinetics. Tungsten oxide-based nanomaterials have received extensive attention due to their great potential in energy-related and environmental catalysis. To maximize the catalytic efficiency of catalysts in practical applications, it is essential to further understand the structure–property relationship of tungsten oxide-based nanomaterials by controlling the surface/interface structure. In this review, recent methods to enhance the catalytic activities of tungsten oxide-based nanomaterials are reviewed, which are classified into four strategies: morphology regulation, phase control, defect engineering, and heterostructure construction. The structure–property relationship of tungsten oxide-based nanomaterials affected by various strategies is discussed with examples. Finally, the development prospects and challenges in tungsten oxide-based nanomaterials are discussed in the conclusion. We believe that this review provides guidance for researchers to develop more promising electrocatalysts for water splitting.

## 1. Introduction

With the rapid development of global modernization, the excessive consumption of non-renewable energy sources, such as oil and coal, has resulted in the crisis of greenhouse effect, energy shortage, and severe environmental pollution [1,2,3,4,5]. It is urgent to develop clean and sustainable energy to alleviate energy pressure and ameliorate environmental problems. Therefore, sustainable clean energy resources, such as wind, solar, tidal, and hydropower, have been extensively studied [6,7,8,9,10,11]. However, these energy sources have the disadvantages of uneven geographical distribution and intermittency, which seriously restrict their popularization and application [12]. Hydrogen fuel is expected to play a significant role in developing sustainable clean energy due to its high energy density, high energy yield (122 kJ/g), and environmentally friendly characteristics [13,14,15]. However, it is estimated that nearly 96% of worldwide hydrogen comes from the conversion of fossil fuels, where the pollution byproducts cause environmental problems, such as climate warming [14,16,17,18,19]. Electrochemical water splitting, as a sustainable method of producing hydrogen with simple operation, mild reaction conditions, environmental protection, and low cost, has attracted much attention from researchers [13,20,21,22]. The water-splitting process involves two half-reactions, i.e., hydrogen evolution reaction (HER) and oxygen evolution reaction (OER). However, due to the slow-kinetic four-electron transfer process with a high reaction barrier, the potential required for water splitting is always higher than the theoretical decomposition potential of water (1.23 V), resulting in additional electrical power consumption [23,24]. Effective electrocatalysts are required to reduce the reaction barrier and accelerate the reaction kinetics. Generally speaking, noble metal (Pt, Ir)-based catalysts show ideal electrocatalytic performance due to their appropriate hydrogen adsorption-free energy. However, the scarcity and high prices of noble metals have prevented their widespread application [25,26]. Therefore, it is essential to develop effective catalysts that are affordable, stable, and low in noble metals.

Among all the candidates, tungsten oxide has attracted much attention because of its abundant reserves, affordable price, and great stability. It is worth noting that tungsten oxides are typical n-type semiconductors with band gaps of 2.6–3.0 V [27], which have a variety of morphologies and crystal phases, and they exhibit reversible surface oxygen ion exchange and excellent redox capacity, owing to their oxidation states flexibly converting between W^6+^, W^5+^, and W^4+^ [28,29,30]. Consequently, tungsten oxide is expected to be a superior substitute for noble metals [29,31]. The fundamental crystal unit of tungsten oxide is WO_6_, an octahedral structure. Tungsten oxide possesses a variety of crystal phases, including tetragonal, orthorhombic, monoclinic I, hexagonal, triclinic, and monoclinic II, due to the varying tilt angles and rotation directions of WO_6_ units [30,32]. Notably, the WO_6_ octahedral units can form shared angles or shared edges, and the more shared angles/edges there are, the more oxygen vacancies and low valence W species are generated, forming a WO_x_ structure with a non-stoichiometric ratio [28,33,34]. The oxygen vacancy is accompanied by the accumulation of free electrons, which transform the semiconductor into a degenerate semiconductor with metallic characteristics, thus improving the conductivity of materials [28,35,36]. Oxygen vacancies also offer unsaturated coordination sites that alter the electron structure, charge density, and conversion in the p-band center of materials, thus improving its inherent electron transport frequency and catalytic performance [29,37,38,39]. Despite these benefits, the Schottky barrier forms at the surface/interface of the electrocatalyst and electrode or electrolyte results in the poor conductivity and the low number of active sites of WO_3_, which hinders the catalytic performance [37,40]. Therefore, the composition and structure of tungsten oxide are designed to modify the electronic structure, improve the conductivity, and increase the number of catalytic active sites [36].

Various effective strategies have been developed to optimize nanostructures of tungsten oxide, thus further enhancing its catalytic properties, as shown in Figure 1. (1) Crystal phase regulation: different crystal phases of the samples have different degrees of interatomic *d*-*sp* orbital hybridization and lattice expansion, thus showing different band gaps and catalytic activities [41,42,43]. Tungsten oxide exhibits tunable catalytic properties due to its adaptable conversion between stoichiometric and non-stoichiometric crystal phases. (2) Morphology regulation: preparing tungsten oxides with various morphologies to expose different surfaces can optimize their adsorption free energy and intrinsic activity, which is of great significance to improve catalytic performances. Additionally, it is shown that the morphology regulation of tungsten oxide can effectively adjust the band gaps and the number of active sites [44]. (3) Defect engineering, including controlling the lattice vacancy and hetero atom doping, has also been used to enhance the intrinsic catalytic activity of tungsten oxide by effectively regulating the electronic structure and increasing the number of active sites [45,46,47]. Numerous experimental and theoretical findings have demonstrated that the introduction of defects to construct tungsten oxide with a non-stoichiometric ratio is more conducive to the adsorption/desorption of intermediate molecules (*H for HER; *OH and *OOH for OER) at the oxygen vacancy and doping sites [35,48,49]. (4) Benefiting from the synergistic effect, the construction of a heterostructure with specific interface configurations has been proved to be an effective way of reducing the reaction barrier of water splitting and promoting the adsorption and desorption of protons [36,50,51,52,53]. In this case, the highly efficient catalytic electrodes possess high electron transport capacity and large active surface area, which might provide sufficient active sites and appropriate adsorption strength. Numerous heterostructures with stable structures, high specific surface areas, and abundant active sites, including semiconductor–WO_x_, WO_x_–C, and metal–WO_x_ heterostructures, have been developed [54,55,56,57]. These fantastic efforts substantially promoted the development of tungsten oxide-based electrocatalysts.

Tungsten oxides are widely utilized in energy storage [58,59,60,61,62,63], sensors [64,65,66,67,68], catalysis [69,70,71,72,73,74], and other fields because of their adjustable valence states (W^4+^, W^5+^, and W^6+^) and band gaps [31,75,76], various morphologies from zero to three dimensions [44,77], and different crystal phases. Here, the research progress of tungsten oxide-based electrocatalysts for water splitting over the last few years is reviewed. Emphatically discussed are the impacts of crystal phase, morphology, defect engineering, and heterojunction effects on the electronic structure and catalytic activity of tungsten oxide-based nanomaterials. We also present a brief summary and outlook of the tungsten oxide catalyst research, with the intention that this review may provide some insights into the construction of high-efficiency oxide catalysts.

## 2. Phase Control

Controlling the crystal phase of tungsten oxide and optimizing its physical and chemical properties have been proven effective in improving catalytic performance [41,42]. The WO_x_ is not simply composed of W^6+^ and O^2−^ ions, which mainly consist of hybrid conduction and valence band states of W 5d and O 2p [32]. The electronic structure of WO_x_ in different crystal phases, such as monoclinic, orthorhombic, and hexagonal phases, is affected by the W–O bond length [32,78,79]. Consequently, it is possible to optimize their catalytic performances by carefully adjusting the crystal phases [30,32,80]. The relatively stable monoclinic and metastable hexagonal phases have attracted extensive research for electrocatalytic performances due to their tunnel structure and rich intercalation [81]. The synthesis methods and the properties (electrolyte, over-potentials at 10 mA cm^−2^, and Tafel slopes) of tungsten oxide-based electrodes with different phases are summarized in Table 1.

Heat treatment is an effective way to control the crystal phase of WO_x_-based nanomaterials, in which the temperature is a vital factor. For instance, Guninel et al. annealed orthotropic WO_3_·H_2_O in air, and they found that the crystal structure transformed to monoclinic WO_3_ with the disappearance of water molecules in the structure. The detailed dehydration process is shown in Figure 2a [83]. Pradhan et al. also have prepared the monoclinic WO_3_ by annealing orthotropic tungsten oxide hydrate at 400 °C in the air (Figure 2b) [82]. It shows that the double-layer capacitance (C_dl_) of monocline WO_3_ is 2.83 times that of the original WO_3_·H_2_O, providing more active surfaces during the catalytic reaction. As a result, the monoclinic WO_3_ exhibits an over-potential of 73 mV at 10 mA cm^−2^ in 0.5 M H_2_SO_4_, which is much lower than that of orthorhombic tungsten oxide hydrate (147 mV). The density functional theory (DFT) results (Figure 2c) proved that the hydrogen proton adsorption energy on P2 1/n monocline WO_3_ (200) is more suitable than that of Pt (111). Halder’s group investigated the effect of heat treatment temperatures on the phase transition. With the increase in calcination temperature, the crystal phase of WO_3_ changed from hexagonal to monoclinic and then to cubic phase [84]. The phase transformation process from hexagonal to monoclinic at 550 °C was further observed by in situ transmission electron microscopy (TEM). The monoclinic WO_3−x_ obtained at this temperature exhibits the best HER activity because of the highest oxygen vacancy concentration.

Certain additives also have an impact on the crystal phase of tungsten oxide during the preparation process. Song’s team precisely prepared the orthorhombic WO_3_·0.33H_2_O and monoclinic WO_3_·2H_2_O by utilizing ethylene diamine tetra acetic acid and DL–malic acid at room temperature, respectively (Figure 2d) [40]. It demonstrated that a lower over-potential (117 mV) and Tafel slope (66.5 mV dec^−1^) of monoclinic WO_3_·2H_2_O were required to reach a current density of 10 mA cm^−2^ in 0.5 M H_2_SO_4_ (Figure 2e). 

Hajiahmadi et al. explored the reaction mechanism and adsorption model of hex-WO_3_ (001) in acid oxygen evolution reaction (OER) (Figure 3) [85]. There are six adsorption models involved: (1) Initial state. The H_2_O adsorbs at the top of the central W site due to the lack of coordination of the central tungsten. (2) Water decomposes to form OH* and H^+^, which are the most stable configurations co-adsorbed at the top W site and adjacent O site, respectively. (3) Dehydrogenation of OH* to form O*. The dissociated hydrogen is adsorbed to another nearby oxygen. (4) The adsorption of water molecule on O*. (5) Dehydrogenation of O*-H_2_O to form OOH*. (6) Dehydrogenation of OOH* to form O_2_.

## 3. Morphology Control

Due to their flexible crystal structures, WO_x_ nanomaterials with rich morphologies exhibit different physical and chemical properties. Reasonable design of catalyst morphology may increase the contact area between catalyst and electrolyte, thus improving the electrocatalytic performance. The impact of morphology on the catalytic performance of WO_x_ has been studied recently, and some advancements have been made [42,86,87]. 

One-dimensional nanostructures, such as nanorods [38] and nanowires [88,89,90], have been extensively studied in tungsten oxide-based nanomaterials. For example, Liang’s group prepared WO_3_ nanowires with rich oxygen vacancy (WO_3_-V_O_ NWs) by hydrothermal method combined with plasma sputtering [89]. The WO_3_-V_O_ NWs grew along the [001] direction (c axis) and exhibited stable electrocatalytic oxygen evolution activity under acidic conditions. Two-dimensional nanostructures have also attracted extensive attention due to their increased surface area, abundant active sites, and appropriate adsorption of intermediates in recent years. Pradhan et al. prepared tungsten oxide hydrate nanoplates with apparent gaps between the stacked nanoplates using the hydrothermal method [82]. Guo’s group prepared hierarchical WO_3_ nanowire arrays on nanosheet arrays (WO_3_ NWA-NSAs) by the hydrothermal method for alkaline OER [29]. The WO_3_ NWA-NSAs electrocatalyst only requires an over-potential of 230 mV to reach the current density of 10 mA cm^−2^ because of its unique hierarchical structure. By changing the composition of surfactants and synthesis parameters of hydrothermal processes, Rajalakshmi et al. prepared various WO_3_ nanomaterials, including one-dimensional nanowires and nanorods, two-dimensional nanoflakes and nanobelts, and three-dimensional nanoparticles, which are star-like and globule-like structures (Figure 4a–g). The influence of morphology on the band gap width and hydrogen evolution performance was also investigated [44]. It was found that the band gaps of tungsten oxide with different morphologies are inconsistent. Among them, WO_3_ nanorods have a higher aspect ratio and better bandgap and adsorption energy in conjunction with the precise cutting of crystal facets along the (001) direction. As a result, the HER performance of one-dimensional WO_3_ nanorods exceeds other morphologies in an acidic electrolyte. For the three-dimensional nanostructure, our group investigated the effect of the etching agent concentration on the morphologies and the alkaline OER performances of Ni–WO_3_ nanostructures [91]. The Ni–WO_3_ octahedral structure (Figure 4h) was in situ etched with (NH_4_)_2_SO_4_, and the serrated Ni–WO_3_ (Figure 4i) was obtained. It was found that the crystal phase of Ni–WO_3_ was unaffected by the concentration of the etching agent, but the serrated Ni–WO_3_ dramatically improves OER performance (an over-potential of only 265 mV at 10 mA cm^−2^) compared with octahedral Ni–WO_3_ (365 mV).

## 4. Defect Engineering

Defect engineering, including oxygen vacancy construction and hetero atom doping, reduces the atomic coordination numbers in the material and, thus, regulates the electronic structure [89,92]. The band gap and the adsorption-free energy of the catalyst could be optimized by regulating the electronic structure of the catalyst, thus reducing the catalytic reaction barrier and improving the reaction kinetics. These doping and defect sites can also potentially increase the number of active sites and the concentration of free carriers, which are essential for reducing the reaction barrier and increasing the electron transfer efficiency [93]. The synthesis methods and the properties (electrolyte, over-potentials at 10 mA cm^−2^, and Tafel slopes) of tungsten oxide-based electrodes modified by defect engineering are summarized in Table 2.

### 4.1. Oxygen Vacancy

Tungsten oxide-based nanomaterials with abundant oxygen defects show great potential for water splitting. Abundant oxygen vacancies can improve the conductivity and promote the adsorption of OH^−^, thus effectively increasing the oxygen evolution activity [89]. For example, Guo’s research group prepared WO_3_ with rich oxygen vacancies by hydrothermal method, and they explored its oxygen evolution properties in the alkaline electrolyte [29]. The abundant oxygen vacancy not only improves the conductivity of WO_3_, but also modifies its electronic structure, so that WO_3_ only needs 230 mV overpotential to reach the current density of 10 mA cm^−2^.

Effect of oxygen vacancy on morphology and properties of WO_x_. Two extra electrons are produced when an O atom is removed from the WO_3_ structure. One or two electrons can be transferred to a neighboring W atom to form the W^5+^ or W^5+^-W^5+^ centers [32]. Additionally, introducing the O-vacancy modifies the splitting of the W-O bond, increases the W–W distance at the O-vacancy position, and narrows the WO_3_ band gap accordingly (Figure 5a–d) [79]. The electrode’s conductivity is improved as a result of abundant oxygen vacancies, which can turn an n-type WO_x_ semiconductor into a degenerate semiconductor with metallic characteristics (Figure 5e) [35,89]. Besides, a high oxygen vacancy concentration will increase the material’s surface roughness and the area in contact with the electrolyte. For example, the surface of tungsten oxide with a smooth hexahedral structure (Figure 5f) turns rough after being annealed in a H_2_ atmosphere, forming a porous structure (Figure 5g) [121]. As shown in Figure 5h,i, the porous WO_2_ HN/NF has a BET surface area, pore size, and specific volume of 22.8 m^2^ g^−1^, 10–100 nm, and 0.138 cm^3^ g^−1^, respectively. Owing to the highly concentrated oxygen vacancies that provide more active sites and narrower band gaps, the porous WO_2_ HN/NF electrode showed lower potential and excellent catalytic stability in HER, OER, and overall water splitting.

Oxygen vacancy recognition. First, the O vacancy can be directly observed by atomic high-resolution TEM (HRTEM), where the variation in atomic column intensity indicates the variation in oxygen atomic occupation. As shown in Figure 6a, the tiny pits shown by the arrows indicate the presence of oxygen vacancies [89]. The variations in intensity and contrast were further highlighted in the colored image and the line profile (Figure 6b). Second, UV-vis diffuse reflectance spectroscopy and UV-vis absorption spectroscopy are also used to identify oxygen vacancy defects. As shown in Figure 6c,d, a stronger photo-response in the visible to infrared region indicates a higher O vacancy concentration in the material [35,40,122]. Besides, electron paramagnetic resonance (EPR), a technique for detecting the chemical environment of unpaired electrons in atoms or molecules, can be used to confirm the existence of oxygen vacancies (Figure 6e). When oxygen vacancies capture electrons, symmetrical EPR signals appear at the position *g* ≈ 2.002. The stronger the intensity of the EPR signal, the higher concentration of oxygen vacancies present in the material [35,89]. Notice that the oxygen vacancy also changes the metal valence in the metal oxide. Therefore, in addition to the direct characterization methods of oxygen vacancies, alternative techniques, such as X-ray photoemission spectroscopy (XPS) and synchrotron X-ray absorption fine structure (XAFS), can also be employed to infer the existence of oxygen vacancies. As shown in Figure 6f, after the oxygen vacancy was introduced, the XPS peaks of W 4f were moved to the lower binding energy region [35,38,122]. For O1s XPS, the peak at ~531.3 eV corresponds to the oxygen vacancy [35,123,124,125]. 

Effect of oxygen vacancy concentration on performance. To further investigate the impact of oxygen vacancy concentration on catalytic performance, Thomas et al. explored the relationship between the electrocatalytic performance of Meso-WO_2.83_ and surface oxidation degree brought on by exposure to air (Figure 6g,h) [35]. With the increase in exposure time, the plasmon resonance of Meso-WO_2.83_ was weakened, accompanied by a red shift, and its electrocatalytic activity gradually decreased, as well. This indicates that abundant oxygen vacancies are advantageous for enhancing catalytic activity. Zeng et al. demonstrated that abundant oxygen vacancies could optimize the hydrogen adsorption Gibbs free energy (ΔG_H*_) using the DFT (Figure 6i) [94].

Method of producing oxygen vacancy. Oxygen vacancies are widespread in transition metal oxides because of their low formation energy [35,39]. Representative methods of producing oxygen vacancies in metal oxides include heat treatment, reductive treatments, and other methods [38,126]. The first method is heat treatment, which involves removing a small amount of lattice oxygen from metal oxides in low-oxygen conditions at high temperatures without causing bulk phase transition. The oxygen vacancy concentration can be adjusted by controlling the heat treatment temperature or the inert gas flow rate. For instance, Halder’s group thermally treated WO_3_ in a vacuum environment to produce WO_3−x_ with rich oxygen vacancies [84]. The second method of producing oxygen vacancies in tungsten oxide is reductive treatments, which create oxygen vacancies with the aid of reducing agents, such as H_2_ [123], NaBH_4_ [127,128], and sodium dodecyl sulfate [29]). Taking the reduction in hydrogen as an example, with the extension of reduction time, WO_3_ gradually evolved into WO_3−x_, WO_2_, and finally, metallic W^0^ [129]. Thomas et al. prepared Meso-WO_2.83_ with oxygen-rich vacancies by replacing bulk materials with mesoporous materials to increase the reduction rate in the hydrogen atmosphere [35]. Due to the slow diffusion of H_2_ molecules in bulk materials, the use of mesoporous WO_3_ with a higher surface area and thin nanoscale pore wall allows H_2_ molecules to diffuse and migrate more easily on its surface and inside. The results show that the mesoporous structure not only dramatically reduces the H_2_ reduction temperature, but also selectively generates WO_2.83_. Other methods of producing oxygen vacancies include plasma treatment, flame, mechanical exfoliation, and hydrothermal methods [29,89,130,131]. For example, as a surface treatment technology, plasma has a certain reduction ability, and the oxygen vacancy generated by it only exists on the surface of the sample [131].

### 4.2. Hetero Atom Doping

Hetero atom doping is also an effective strategy to prepare tungsten oxide-based nanomaterials with abundant defects, which achieve a significant leap in catalytic performance by regulating the electronic coordination environment, the number of active sites, and the adsorption strength of intermediates [48,113]. In practical applications, atom-doped materials usually contain both hetero atoms and oxygen vacancies, which can cooperatively promote the catalytic activity [93,118,132]. The effect of doping on catalytic performance can be adjusted by changing the type [48,105] and concentration [115,118,133] of hetero atoms. Atom doping could be divided into noble metal atomic doping and non-noble metal atomic doping. For the former, the noble metal atoms typically exist as single atoms or atomic clusters to reduce the costs and improve the utilization rate of noble metal atoms [112,113,134,135,136]. For example, Hou et al. reported Pt sing-atoms supported on monolayer WO_3_ (Pt-SA/ML-WO_3_) for HER [104]. The high-angle annular dark-field scanning transmission electron microscopy (HAADF-STEM) images (Figure 7a,b) and ICP result showed that 0.20 wt% Pt atoms were immobilized on ML-WO_3_. HRTEM images demonstrate the existence of defects, including lattice distortion, as well as W and O vacancies (Figure 7c,d). The HER performance of Pt-SA/ML-WO_3_ (over-potential of 22 mV at 10 mA cm^−2^, Tafel slope of 27 mV dec^−1^) was even better than that of 20% Pt/C (over-potential of 34 mV at 10 mA cm^−2^, Tafel slope of 28 mV dec^−1^). The improved performance is mainly attributed to the strong interaction between Pt single atoms and ML-WO_3_, which drastically tunes the electronic structure of the catalyst, endowing Pt-SA/ML-WO_3_ with a strong conductivity and an adequate ∆G_H*_ (Figure 7e). Sun et al. investigated the overall water-splitting performance of the iridium-doped tungsten trioxide array (Ir-doped WO_3_) in acidic conditions [100]. Ir can preempt some of the electrons in the WO_3_ matrix to optimize its electronic structure because Ir^4+^ has a smaller radius and a higher atomic number than W^6+^. The Ir-doped WO_3_ exhibited low cell voltages of 1.56 and 1.68 V to drive the current densities of 10 and 100 mA cm^−2^, respectively. Ma’s group analyzed the effects of oxygen vacancy and Ru doping on electronic states and *d*-band center of WO_3_ by density functional theory (DFT) calculations [136]. As shown in Figure 7f, when only oxygen vacancies exist, the electrons produced by oxygen vacancies are transferred to neighboring W atoms. After doping Ru atom, electrons are mainly transferred from Vo site to adjacent Ru site, and a few electrons are transferred to an adjacent W site. Therefore, Ru sites with sufficient electrons in Vo-WO_3_/Ru SAs may be active centers for adsorption intermediates in acidic media. In alkaline and neutral solutions, the oxygen atoms in water molecules are more easily captured by the electron-deficient W site, and the generated H* migrates to the nearby Ru site to form H_2_. In addition, the *d*-band center of Vo-WO_3_/Ru SAs (−5.180 eV) is much lower than that of WO_3_ (−3.252 eV) and Vo-WO_3_ (−4.133 eV), indicating that the strength of H* bond is weakened (Figure 7g), which is conducive to the desorption of H* from the surface and the promotion of HER.

In addition to doping the noble metal atoms, doping of non-noble metal atoms (Co, Fe, Ni, F, and Mo) could also regulate the physicochemical properties of the catalyst and effectively improve the catalytic performance of tungsten oxide-based nanomaterials [48,107,111,115,116,133,137,138]. For example, Xiao et al. simulated the *d*-orbital hybridization of WO_2_ by a series of transition metal heteroatoms using first principles and explored HER properties of M-WO_2_ (M = Fe, Co, Ni, and Cu atoms) [105]. The Fe, Co, Ni, or Cu heteroatoms replace one of the W atoms (W-M bond) to form higher *d*-band packed atomic coordination, resulting in increased filling of W–M bond antibonding orbitals and weakening of bond strength (Figure 8a). The DFT result showed that the Ni–W site has modest hydrogen adsorption (ΔG_H*_ = −0.43 eV) due to the dynamic transfer of some bonded electrons from the W-W/Ni bonds to the Ni-O bonds after the substitution of W atoms by Ni, which reduces the free energy, weakens the metal–metal bond, and increases the bond length of W-W/Ni (Figure 8b). Therefore, Ni-WO_2_ only needs 83 mV to reach the current density of 10 mA cm^−2^ in an alkaline environment. To investigate the effect of Ni atom doping on the localized structure of tungsten oxide, the author compared the W L_3_-side X-ray absorption near side structures (XANES) and further proved that the addition of Ni atoms can effectively reduce the d-band occupation state on W atoms. Comparing the whole contour plots of wavelet transform (WT) (Figure 8c,d) and the charge density difference slices (Figure 8e,f) of WO_2_ and Ni-WO_2_, it was found that the R-value at about k = 12.2 Å^−1^ in the WT spectrum of Ni-WO_2_ is increased, and the charge density between Ni–W is significantly decreased. All these results indicate that Ni doping can effectively regulate the coordination strength between metal atoms and reduce the ability to capture electrons of W atoms, which is closely related to the absorption/desorption behavior on the catalyst surface.

## 5. Heterostructure Construction

Constructing heterostructured nanomaterials composed of tungsten oxide and another material is an effective strategy to improve the catalytic performance of tungsten oxide [139,140,141]. The heterostructured nanomaterials reaches the catalytic performance of 1 + 1 > 2 by exposing more active sites and promoting interfacial electron transfer, which is called the “spillover” mechanism of the system [52,141]. The synthesis methods and the catalytic properties of tungsten oxide-based heterostructure electrodes with different morphologies (such as nanowires, nanoflakes, nanospheres, urchin-like, and so on) are summarized in Table 3. Here, tungsten-oxide heterostructures are divided into three types according to the different components: (1) semiconductor–WO_x_; (2) WO_x_–C; and (3) metal–WO_x_. 

### 5.1. Semiconductor–WO_x_

The interaction between tungsten oxide and another semiconductor is conducive to adjusting its electronic structure. When tungsten oxide comes into contact with a semiconductor with different Fermi levels, the electrons will spontaneously diffuse from the semiconductor with a high Fermi level to another component until the chemical potential of the two parts is equal, thus forming a semiconductor–WO_x_ heterojunction structure [172]. Consequently, net charges accumulate at the contacting interface, which lowers the initially higher Fermi level and raises the initial lower Fermi level. Meanwhile, the electronic band of the contacting semiconductor bends over subject to the movement of Fermi levels, generating different types of band alignments. Due to the synergistic effect and electronic effect between the components, the catalytic performance of the composites is improved. Wang and his colleagues prepared the Ni_2_P–WO_3_ nanoneedle structure on carbon cloth using a combination of in situ electrodeposition and phosphating treatment methods [151]. The XPS results demonstrate the electrons transfer from Ni to P in Ni_2_P–WO_3_. Benefiting from the heterojunction structure, Ni_2_P–WO_3_ exhibits excellent HER catalytic performance in both acidic (over-potential of 107 mV at a current density of 10 mA cm^−2^) and alkaline (over-potential of 105 mV at a current density of 10 mA cm^−2^) environments. Peng et al. designed a Fe_2_P–WO_2.92_ heterostructure on nickel foam by a facile consecutive three-step synthesis method [72]. The oxygen vacancies and the synergistic effect between WO_2.92_ and Fe_2_P facilitated a drastic reduction in over-potential for the catalytic OER performance of Fe_2_P–WO_2.92_/NF (over-potential of 215 mV at 10 mA cm^−2^ in 1.0 M KOH solution). 

Moreover, the interfacial richness of the two phases in the semiconductor–WO_x_ heterostructure directly affects the number of active sites. Yang’s group prepared Ni_17_W_3_/WO_2_ heterojunction on nickel foam (WO_2_/Ni_17_W_3_/NF) by the hydrothermal and annealing method (Figure 9a) [150]. WO_2_/Ni_17_W_3_ heterojunctions increase the exposure of active edge sites and facilitate the water dissociation and H intermediates association during HER kinetics. Therefore, WO_2_/Ni_17_W_3_/NF demonstrated high catalytic efficiency for HER with a low over-potential of 35 mV at 10 mA cm^−2^. Following this work, Liu’s group prepared R-Ni_17_W_3_/WO_2_ catalysts on Ni foam and explored the effect of Ni_17_W_3_ particle size decorated on the NiWO_4_/WO_2_ substrate for hydrogen evolution reaction (Figure 9b) [152]. The R-Ni_17_W_3_/WO_2_ with larger Ni_17_W_3_ particles exhibited superior HER catalytic activity (over-potential of 48 mV at 10 mA cm^−2^), resulting from more interfaces and more active sites (Figure 9c). For R-Ni_17_W_3_/WO_2_/NiWO_4_, H_2_O adsorption/deionization occurred at the Ni_17_W_3_/NiWO_4_ interface, and H_ads_ and OH_ads_ were adsorbed on Ni_17_W_3_ and NiWO_4_, respectively. The H_ads_ on Ni_17_W_3_ overflow to WO_2_ preferentially in Ni_17_W_3_/WO_2_ interface, which is conducive to the desorption of H_2_ (Figure 9d). Our group also prepared a TA-Fe@Ni–WO_x_ hierarchical structure by the interfacial coordination assembly process. After the introduction of the TA–Fe layer, the electrons transfer from W and Ni to TA–Fe, and as a result, the TA–Fe@Ni–WO_x_ has an upward-moving Fermi energy, a smaller ionization potential, and a more electron-rich environment, which is more conducive to OER [167].

### 5.2. WO_x_–C

Creating a heterostructure with another conductive material is a typical way to increase the electrocatalyst’s overall electronic conductivity. Carbon materials, including graphene oxide (GO), carbon nanotube (CNT), carbon paper, and carbon cloth, are often used due to their superior electronic conductivity, high specific surface area, and excellent chemical durability [56,173,174,175,176]. WO_x_/carbon composites have been widely used in electrocatalytic water splitting [71,142,143,145,146,177]. In this case, carbon-encapsulated WO_x_ with rich oxygen vacancies was synthesized by pyrolyzing carbon/tungsten mixture (Figure 10a,b) [95,144,178,179], which has a favorable impact on enhancing charge transfer and compensating for the weak hydrogen adsorption of the tungsten oxide. The effects of W content [95], annealing time [179], and annealing temperature [178] on HER performance of WO_x_/C have been deeply explored. For example, Yin et al. studied the effects of different annealing times and temperatures on hydrogen evolution properties of WO_x_/C in 0.5 M H_2_SO_4_ (Figure 10c) [179]. Pan’s group introduced 15 nm thick carbon-based shells on the surface of tungsten oxide nanospheres (CTO) and investigated its catalytic performance in alkaline OER [145]. It was found that the overpotential of nanoparticles at 50 mA cm^−2^ decreased from 360 to 317 mV after the introduction of a carbon-based shell (Figure 10d). The improved catalytic performance is attributed to the carbon-based shell that speeds up the electron transfer between the catalyst and the reactant, provides the catalytic active site, and promotes the adsorption of the catalyst to the reactant and the dissociation of the O–H bond.

### 5.3. Metal–WO_x_

Conductive metal is also widely used to increase the electrocatalyst’s overall electronic conductivity and adjust the electronic structure of WO_x_. In order to reduce the cost of the catalyst, the metal part in metal–WO_x_ heterostructure often exists in the form of single atoms, clusters, or nanoparticles [71,120,156]. Li et al. prepared Pt@WO_3_/C with three-dimensional nano architectures for HER via the water–oil two-phase microemulsion method and the annealing treatment [156]. Pt nanoparticles with a diameter of about 4 nm were monodispersed on the surface of WO_3_/C structures (Figure 11a,b). The over-potential at 10 mA cm^−2^ of Pt@WO_3_/C as HER electrocatalyst was 149 mV (Figure 11c), which was smaller than that of WO_3_/C (244 mV). The mechanism of HER consists of three main steps, including H_2_O adsorption, H_2_O dissociation, and H^*^ desorption (Figure 11d). The corresponding free energy calculation results show that the introduction of Pt has no obvious effect on water absorption, but it can promote the water dissociation and H^*^ desorption of WO_3_, thus accelerating the HER process (Figure 11e). The XPS results (Figure 11f) also confirm that the peaks of W 4f were positively shifted after the introduction of Pt due to the difference in electronegativity between W and Pt atoms, resulting in the transfer of electrons from W to Pt. In another work, Pt-WO_3−x_ nanodots were anchored on rGO for water splitting [37]. The optimized composite Pt-WO_3−x_@rGO exhibited the highest HER, OER, and overall water-splitting activities in alkaline environments (over-potentials of about 34 mV, 174 mV, and 1.55 V at 10 mA cm^−2^, respectively). At the same time, its overall water-splitting performance showed excellent durability at 1.55 V, where 93.3% initial potential could be maintained after 14 h of cycling. 

## 6. Summary and Outlook

In this review, we focus on the recent research progress of tungsten oxide-based nanomaterials for water splitting, especially revealing the mechanism of structural design and component regulation to improve the electrocatalytic performances, which is expected to guide the preparation of high-performance tungsten oxide-based electrocatalysts. The practical strategies for improving the performance of tungsten-oxide catalysts are discussed in this review. The band gap and contact area of a tungsten oxide-based electrode can be adjusted by controlling the morphology and crystal phase, thus improving its catalytic performance. The electronic structure of WO_x_ can also be optimized through defect engineering to provide more active sites and an excellent electronic environment for catalytic reactions. Additionally, the synergistic effect can boost the electron transfer of tungsten oxide and provide more active boundaries, which is another commonly used effective strategy for enhancing catalytic performance. Due to the adjustment of structure and composition, noble metal single atom/tungsten oxide/carbon, the best tungsten-based catalyst at present, greatly reduces the cost on the premise of excellent catalytic performance. Despite the advancement of these methods, the practical application of tungsten-based materials in water splitting is still in the early stage, and there are still many problems and challenges.

First, defect engineering has been widely used in regulating the electronic structure of tungsten oxide and promoting its catalytic performance. However, the precise control of the defect location and concentration is difficult, and the characterization technology of defect types and concentration is insufficient. Therefore, the improvement of catalytic performance by defect engineering lacks an intrinsic understanding. The development and utilization of more advanced in situ preparation and characterization technologies may somewhat solve this problem. For example, in situ atmospheric spherical correction transmission electron microscopy can be used to directly observe the defect formation process in nanomaterials and the resulting crystal phase changes.

Besides, the current research on the structure–property relationship mainly adopts the post-analysis method, which compares the electrocatalyst’s structure before and after the catalytic reaction and infers the structure evolution during the reaction based on the static characterization results. However, the structure of tungsten oxide dynamically evolves in the catalytic reaction process. The morphology, chemical composition, and electronic structure of tungsten oxide are constantly changing and recycling. As a result, the structure differences before and after long-lasting catalytic reactions may not reflect the real active sites. To further understand the structure–property mechanism of tungsten oxide-based electrocatalysts, it is necessary to develop and utilize in situ characterization methods, such as in situ electron microscope and the in situ XRD technique, to record the dynamic structural changes of the catalysts in real-time during the reaction. 

We believe that the growth mechanism of material preparation and structure–property relationship can be better understood by in situ observation of the structure and defect location and concentration, as well as in situ characterization of catalytic processes. Combined with advanced theoretical simulation techniques, it is possible to obtain tungsten oxide-based electrocatalysts with high intrinsic activity. All these developments will significantly contribute to the rapid development of renewable energy storage and conversion.

## Figures and Tables

**Figure 1 nanomaterials-13-01727-f001:**
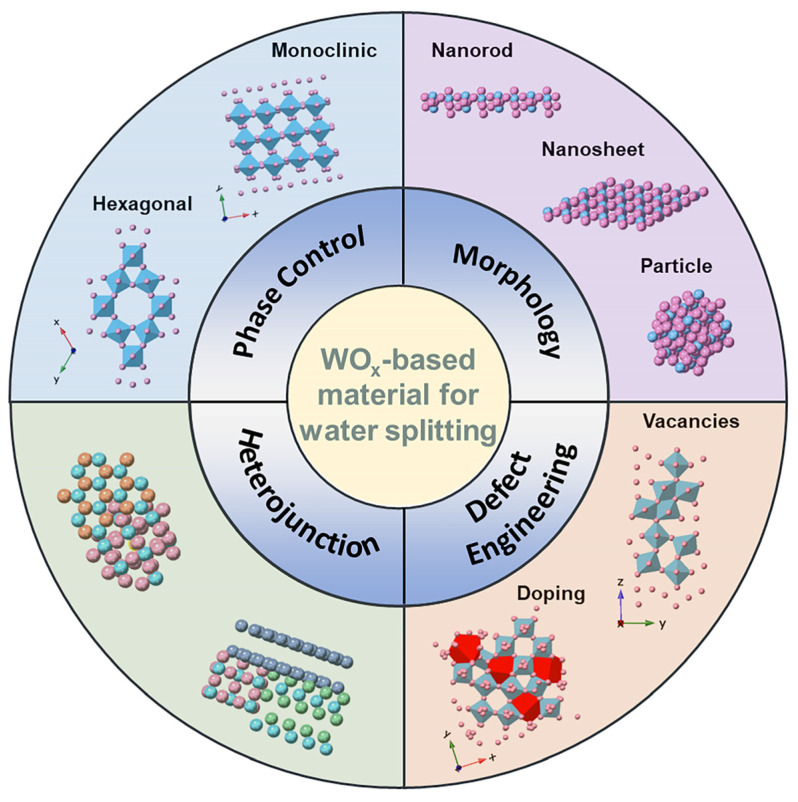
Strategies for improving the electrocatalytic performances of WO_x_-based electrocatalysts.

**Figure 2 nanomaterials-13-01727-f002:**
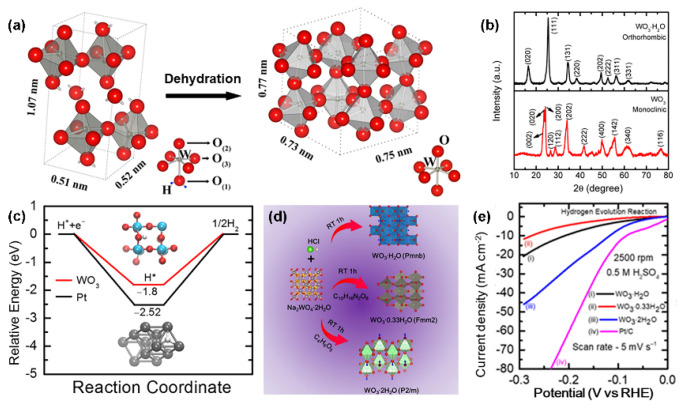
(**a**) Schematic of the orthorhombic unit cell of tungstite, which converts to the monoclinic tungsten oxide unit cell by dehydration at elevated temperatures (≥300 °C). Reprinted with permission from [83], with permission from Springer, 2014. (**b**) XRD patterns of orthorhombic WO_3_·H_2_O and monoclinic WO_3_. (**c**) Calculated energy landscapes of the HER on WO_3_ (200) and Pt (111). Reprinted with permission from [82], with permission from American Chemical Society, 2017. (**d**) Synthesis process and (**e**) linear sweep voltammograms at 2500 of hydrated tungsten oxide (WO_3_·nH_2_O, n values 0.33, 1.00, or 2.00) at room temperature. Reprinted with permission from [40], with permission from American Chemical Society, 2020.

**Figure 3 nanomaterials-13-01727-f003:**
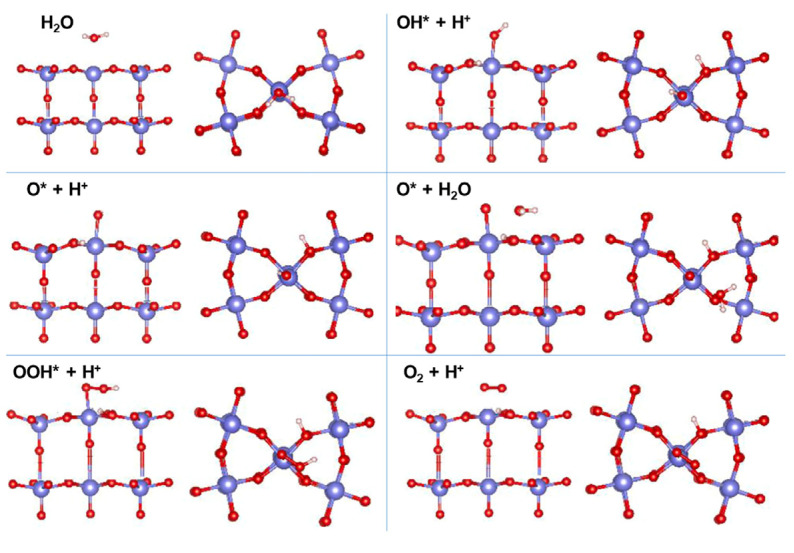
Optimized geometries of hex-WO_3_ at each step in the acid oxygen evolution reaction, top and side view. O: red, W: purple, H: white. Reprinted with permission from [85], with permission from Elsevier, 2022.

**Figure 4 nanomaterials-13-01727-f004:**
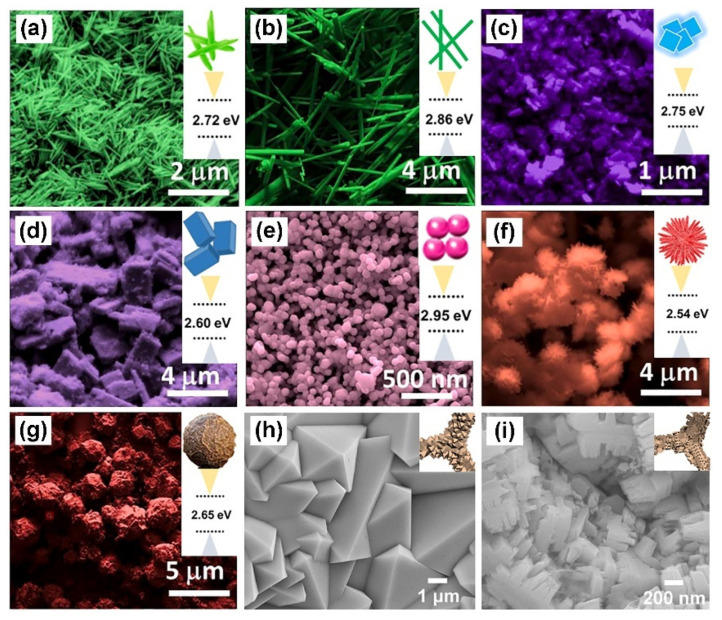
FESEM image and of bandgap (**a**) WO_3_ nanowires (W–NW), (**b**) WO_3_ nanorods (W–NR), (**c**) WO_3_ nanoflakes (W–NF), (**d**) WO_3_ nanobelts (W–NB), (**e**) WO_3_ nanoparticles (W–NP) of W–NP, (**f**) WO_3_ star-like structures (W–S), and (**g**) WO_3_ globule-like structures (W–G). Reprinted with permission from [44], with permission from Elsevier, 2022. FESEM image of octahedron-like (**h**,**i**) serrated Ni–WO_3_. Reprinted with permission from [91], with permission from Elsevier, 2022.

**Figure 5 nanomaterials-13-01727-f005:**
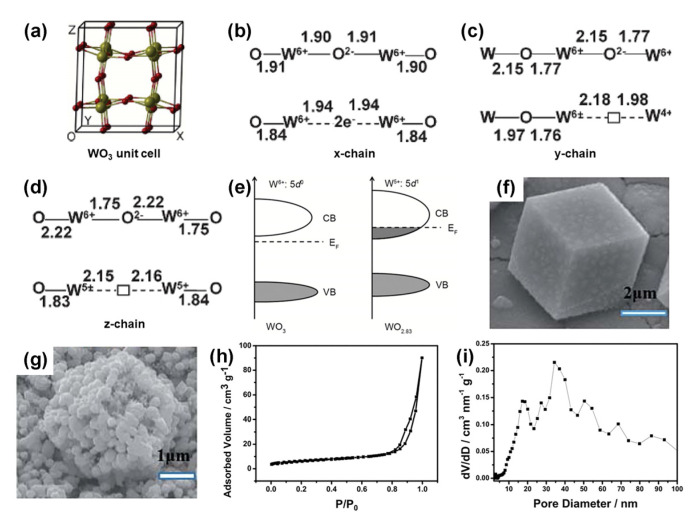
(**a**) Ball-and-stick model of RT monoclinic WO_3_ unit cell with 32-atom or 64-atom. The large green (gray) and small red (dark) balls represent the W and O atoms, respectively. The *x* (**b**), *y* (**c**), and *z* (**d**) direction -W-O-W- chains with (WO_3−x_, x = 0.12 for 32-atom, x = 0.06 for 64-atom) and without (WO_3_) oxygen vacancy. Distances are in Å. The two excess electrons are created by removing a neutral O atom from WO_3_, which can be either paired up in a singlet (closed shell) spin state or unpaired in a triplet spin state. Reprinted with permission from [79], with permission from the American Physical Society, 2011. (**e**) Schematic illustration of the band structures of Meso-WO_2.83_ and Meso-WO_3_. Reprinted with permission from [35], with permission from the Royal Society of Chemistry, 2018. SEM of the WO_2_ hexahedral networks supported on nickel foam before (**f**) and after (**g**) calcination. (**h**) N_2_ adsorption-desorption isotherms of porous WO_2_ HN/NF and (**i**) corresponding pore size distribution. Reprinted with permission from [121], with permission from the Royal Society of Chemistry, 2017.

**Figure 6 nanomaterials-13-01727-f006:**
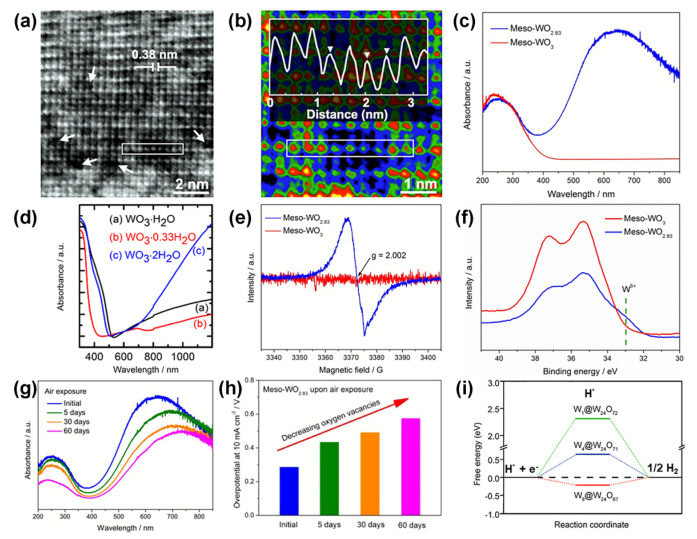
(**a**) TEM images (Numerous small pits shown by the arrows provide more coordinated unsaturated sites for OER electrocatalysis) and (**b**) colored atomic-resolution TEM image of WO_3_-V_o_ with a line profile of nine atoms (indicated by a box), the differences in intensity and contrast are further highlighted in the colored image and the line profile. Reprinted with permission from [89], with permission from the Chinese Chemical Society, 2020. (**c**) UV/vis diffuse reflectance spectra of Meso-WO_3_ and Meso-WO_2.83_. Reprinted with permission from [35], with permission from the Royal Society of Chemistry, 2018. (**d**) UV-vis absorption spectra of WO_3_·H_2_O, WO_3_·0.33H_2_O, and WO_3_·2H_2_O. Reprinted with permission from [40], with permission from the American Chemical Society, 2020. (**e**) EPR spectra at room and (**f**) XPS of Meso-WO_3_ and Meso-WO_2.83_. (**g**) UV/vis diffuse reflectance spectra and (**h**) the overpotential comparison at the current density of 10 mA cm^−2^ of the Meso-WO_2.83_ products upon air exposure time. Reprinted with permission from [35], with permission from the Royal Society of Chemistry, 2018. (**i**) Free-energy diagram for hydrogen adsorption at the W site on the WO_3_ (010) slab with different O vacancies. Reprinted with permission from [94], with permission from the American Chemical Society, 2017.

**Figure 7 nanomaterials-13-01727-f007:**
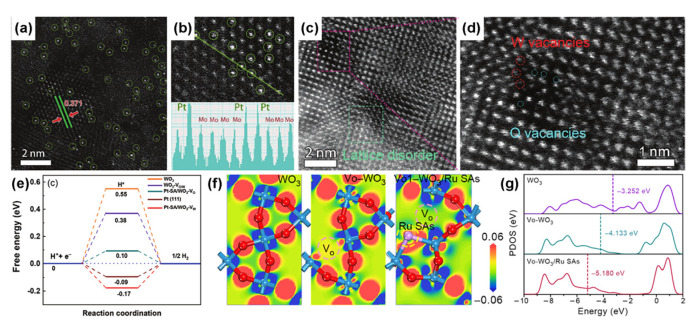
(**a**) HAADF-STEM image, (**b**) magnified HADDF-STEM image and corresponding intensity profiles of the line section, and (**c**,**d**) magnified HADDF-STEM images of Pt-SA/ML-WO_3_. (**e**) Calculated free energy profiles of HER at the equilibrium potential for WO_3_, WO_3_-V_O/W_, Pt-SA/WO_3_-V_O_, Pt-SA/WO_3_-V_W_, and metal Pt (111). Reprinted with permission from [104], with permission from Wiley, 2021. (**f**) Electron density difference isosurfaces (where red and blue areas indicate the charge accumulation and depletion, respectively) and (**g**) partial electronic density of states of WO_3_, Vo-WO_3_, and Vo-WO_3_/Ru SAs. Reprinted with permission from [136], with permission from Elsevier, 2022.

**Figure 8 nanomaterials-13-01727-f008:**
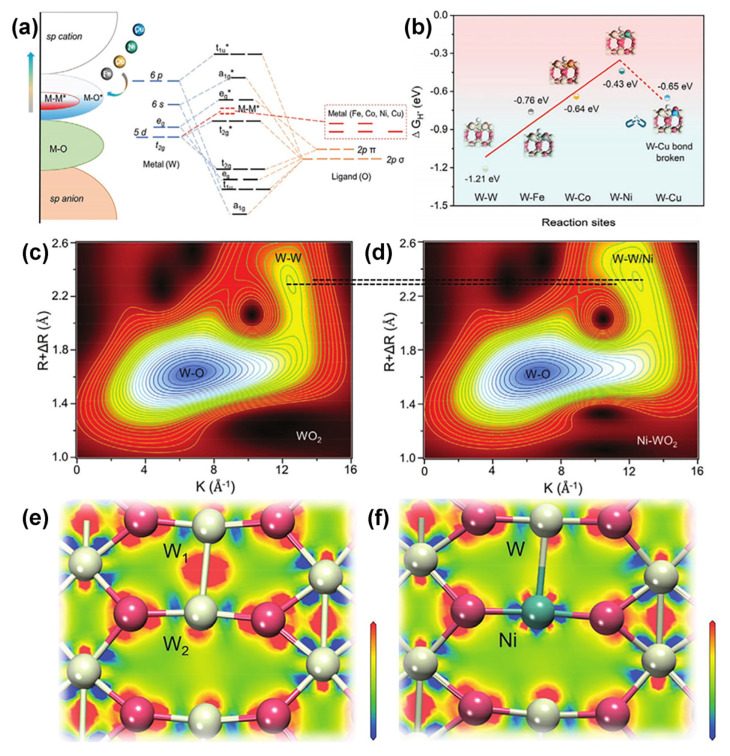
(**a**) Schematic illustration of the W 5d-M 3d orbital modulation of M–WO_2_ (M = Fe, Co, Ni, Cu, etc.). (**b**) The calculated hydrogen adsorption Gibbs free energy on different reaction sites of M–WO_2_ (011). The whole contour plots of wavelet transform (WT) of (**c**) WO_2_ and (**d**) Ni–WO_2_. The top-view electron density difference of (**e**) WO_2_ (011) and (**f**) Ni–WO_2_ (011), ranging from −0.1 to 0.1 e Å^−3^. Reprinted with permission from [105], with permission from Wiley, 2022.

**Figure 9 nanomaterials-13-01727-f009:**
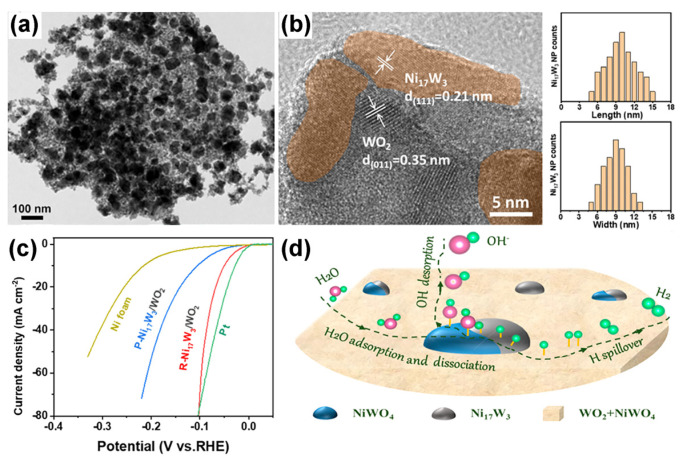
(**a**) TEM image of the Ni_17_W_3_/WO_2_. Reprinted with permission from Reprinted with permission from [150], with permission from the American Chemical Society, 2019. (**b**) HRTEM image and size distribution plots for R-Ni_17_W_3_/WO_2_ nanoparticles. (**c**) Polarization curves for Pt foil, Ni foam, P-Ni_17_W_3_/WO_2_, and R-Ni_17_W_3_/WO_2_. (**d**) Schematic diagram of R-Ni_17_W_3_/WO_2_ nanoparticles accelerating the rate of hydrogen evolution reaction. Reprinted with permission from [152], with permission from the American Chemical Society, 2021.

**Figure 10 nanomaterials-13-01727-f010:**
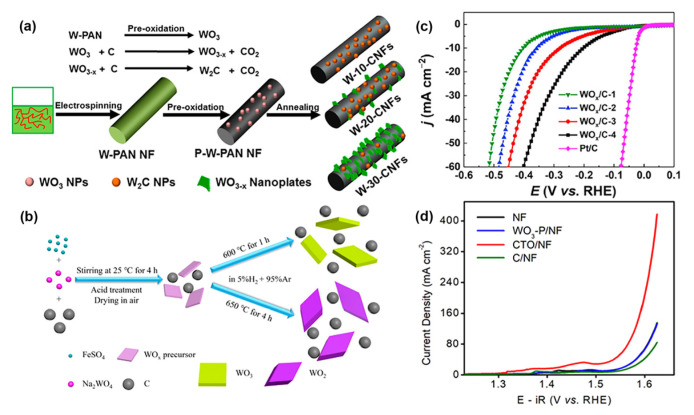
(**a**) Schematic diagram of WO_3−x_ synthesis. Reprinted with permission from [95], with permission from the American Chemical Society, 2016. (**b**) Schematic diagram of WO_x_/C synthesis steps. (**c**) Polarization curves with *iR* compensation of the WO_x_/C in 0.5 M H_2_SO_4_ aqueous solution. Reprinted with permission from [179], with permission from Elsevier, 2021. (**d**) LSV polarization curves measured at a scan rate of 5 mV s^−1^ in 1 M NaOH. Reprinted with permission from [145], with permission from the American Chemical Society, 2020.

**Figure 11 nanomaterials-13-01727-f011:**
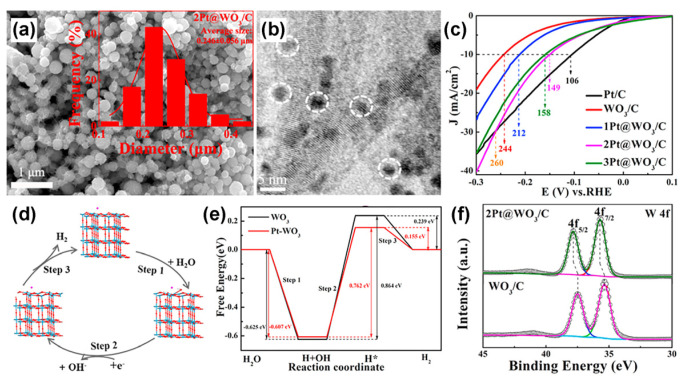
(**a**) FESEM and (**b**) HRTEM images of 2Pt@WO_3_/C. Insert of (**a**) is the size distribution. The white circle in (**b**) are Pt particles. (**c**) The polarization curves of WO_3_/C, 1Pt@WO_3_/C, 2Pt@WO_3_/C, and 3Pt@WO_3_/C. (**d**) the HER process and corresponding chemisorption models of Pt-WO_3_ in neutral media, and (**e**) the corresponding free energy level diagram. (**f**) XPS of WO_3_/C and 2Pt@WO_3_/C. Reprinted with permission from [156], with permission from Elsevier, 2022.

**Table 1 nanomaterials-13-01727-t001:** Application of tungsten oxide-based catalyst with different phases in water splitting.

Electrocatalyst	Synthesis Method	Electrolyte	Application	Scanning Speed (mV s^−1^)	Overpotential at 10 mA cm^−2^ (mV)	Tafel Slope (mV dec^−1^)	Ref.
Monoclinic WO_3_·2H_2_O	Wet-chemical route	0.5 M H_2_SO_4_	HER	5	117	66.5	[40]
Orthorhombic WO_3_·H_2_O					209	198	
Orthorhombic WO_3_·0.33H_2_O					276	376.5	
Hexagonal-WO_3_	Hydrothermal method	0.5 M H_2_SO_4_	HER	2	83	48	[42]
Monoclinic-WO_3_					106	78	
m-WO_3_	Hydrothermal method, thermal treatment	0.5 M H_2_SO_4_	HER	5	168	83	[41]
h-WO_3_					257	157	
Monoclinic WO_3_	In ethanol and kept under ambient temperature, thermal treatment	0.5 M H_2_SO_4_	HER	1	73	39.5	[82]
Orthorhombic WO_3_·H_2_O					147	43.9	

**Table 2 nanomaterials-13-01727-t002:** Application of tungsten oxide-based catalyst modified by defect engineering in water splitting.

Electrocatalyst	Synthesis Method	Electrolyte	Application	Scanning Speed (mV s^−1^)	Overpotential at 10 mA cm^−2^ (mV)	Tafel Slope (mV dec^−1^)	Ref.
WO_3−r_	Liquid exfoliation	0.5 M H_2_SO_4_	HER	2	38	38	[94]
WO_3−x_	Electrospinning	0.5 M H_2_SO_4_	HER	5	185	89	[95]
Meso-WO_2.83_	H_2_ reduction	0.5 M H_2_SO_4_	HER	5	287	95	[35]
WO_2.9_	Wet grinding method, thermally treated	0.5 M H_2_SO_4_	HER	5	70	50	[96]
Oxygen vacancies-rich tungsten oxides	Hydrothermal method, thermal treatment	1 M KOH	HER	5	25	49.25	[97]
W_18_O_49_	Microwave–solvothermal treatment	0.5 M H_2_SO_4_	HER	5	54	30	[98]
		0.1 M Na_2_SO_4_			~200	80	
Ta-doped WO_3_	Hydrothermal method	1 M H_2_SO_4_	HER	2	480	65	[99]
Ir-doped WO_3_	Hydrothermal method, thermal treatment	0.5 M H_2_SO_4_	OER	1	258	48.9	[100]
			HER		36	92	
			OWS		1.56	-	
W/WO_2_	Hydrothermal method, thermal treatment	0.5 M H_2_SO_4_	HER	5	297	74.5	[101]
4% Sm doped WO_3_	Hydrothermal method	0.5 M H_2_SO_4_	HER	5	54	74.5	[102]
5% Sm doped WO_3_			OER		90	138	
4% Sm doped WO_3_ ||5% Sm doped WO_3_			OWS		1.6 V	-	
Ag-WO_3_	Sonochemical method	0.5 M H_2_SO_4_	HER	2	207	52.4	[103]
Pt-SA/ML-WO_3_	Space confined strategy, thermal treatment	0.5 M H_2_SO_4_	HER	2	22	27	[104]
Ni-WO_2_/CP	Hydrothermal method, thermal treatment	1 M KOH	HER	5	83	79	[105]
Ni-WO_2_/NF					41	47	
Fe-P/WO_2_	Organic–inorganic hybridization method	0.5 M H_2_SO_4_	HER	5	48	47	[106]
Ni_0.19_WO_4_	Hydrothermal method, thermal treatment	1 M KOH	OER		240	47	[107]
			HER		200	78	
			OWS		1.59 V	-	
Co-WO_2.7−x_	Hydrothermal method, thermal treatment	1 M KOH	HER	5	59	86	[48]
Ni/WO_x_	Hydrothermal method, thermal treatment	1 M KOH	HER	5	42	57.9	[108]
			OER		395.7@100 mA cm^−2^	100	
			OWS		1.52	-	
Ni-doped W_18_O_49_	Hydrothermal method, thermal treatment	1 M KOH	HER	2	90	92	[93]
			OER		240@20 mA cm^−2^	106	
			OWS		1.56 V	-	
10 wt% Ir/W_18_O_49_ nanowire	Solvothermalsonication and dispersion, thermal treatment	0.5 M H_2_SO_4_	HER	5	41	38	[109]
		1 M PBS			83	66	
Fe-WO_x_	Hydrothermal	1 M KOH	OER	-	380	51.7	[110]
Ni_0.78_WO_2.72_	Wet chemical,thermal treatment	0.1 M KOH	OER	10	270	-	[111]
Pt SA/WO_3−x_	Incipient wetness impregnation method	0.5 M H_2_SO_4_	HER	5	47	45	[112]
Ru-WO_2.72_	Hydrothermal method, self reduction	0.5 M H_2_SO_4_	HER	5	40	50	[113]
Ag/WO_3−x_	Situ reduction method	0.5 M H_2_SO_4_	HER	5	30	40	[114]
Mo-W_18_O_49_	Hydrothermal method	0.5 M H_2_SO_4_	HER	5	45	54	[115]
NiFe-W_18_O_49_	Hydrothermal method	0.1 M KOH	OER	5	325	42	[116]
5 at% Pd doped W_18_O_49_	Solvothermal method	0.5 M H_2_SO_4_	HER	1	137	35	[117]
1 at% Mo incorporated W_18_O_49_ nanofibers	Solvothermal method	0.5 M H_2_SO_4_	HER	1	262	49	[118]
Co_0.5_Fe_0.5_WO_4_	A polyol route	1 M KOH	OER	1	360	36.3	[119]
Pt/def-WO_3_	Hydrothermal method and deposition–precipitation method	0.5 M H_2_SO_4_	HER	1	42	61	[120]

**Table 3 nanomaterials-13-01727-t003:** Application of tungsten oxide-based heterostructured catalysts in water splitting.

Electrocatalyst	Synthesis Method	Electrolyte	Application	Scanning Speed (mV s^−1^)	Overpotential at 10 mA cm^−2^ (mV)	Tafel Slope (mV dec^−1^)	Ref.
WO_2_-C nanowires	Calcination	0.5 M H_2_SO_4_	HER	2	58	46	[142]
WO_3_ nanoflakes/B-AC	Sonochemical method	1 M KOH	OER	1	320	48	[143]
			HER		360	14	
WO_3_@NPRGO	Redox reaction and carbonization	0.5 M H_2_SO_4_	HER	5	225	87	[144]
Carbon-based shell coated tungsten oxide nanospheres	Hydrothermal method, thermal treatment	1 M NaOH	OER	5	317@50 mA cm^−2^	70	[145]
WO_x_ NWs/N-rGO	Hydrothermal method, thermal treatment	0.5 M H_2_SO_4_	HER	2	40	38.2	[146]
WO_x_/C nanowires	Thermal treatment	0.5 M H_2_SO_4_	HER	2	108	46	[147]
Urchin-like CC@WO_3_/Ru SA-450	Hydrothermal–calcining–galvanostatic deposition	0.5 M H_2_SO_4_	HER	2	17	54.7	[136]
		1 M KOH			34	57.5	
		1 M PBS			64	91.9	
NiWO_4_/WO_3_ fibers	Electrospinning schematic	0.5 M H_2_SO_4_	HER	1 mA/cm^2^	80	50.27	[132]
		0.1 M KOH			60	41.97	
IrO_2_/WO_3_	Wet-chemical route	0.1 M HClO_4_	OER	10	-	65	[50]
WO_3_-TiO_2_ particles	Acid catalyzed peptization method	32% of NaOH	HER	-	120	98	[148]
WS_2_/WO_3_ nanosheets	Hydrothermal method	0.5 M H_2_SO_4_	HER	2	395	50	[149]
WO_3_/Ni_3_S_2_	Hydrothermal method, thermal treatment	1 M KOH	HER	5	249@100 mA cm^−2^	45.06	[51]
Ni_17_W_3_/WO_2_ nanoparticles	Hydrothermal method, thermal treatment	1 M KOH	HER	5	35	41.6	[150]
W_18_O_49_/NiWO_4_	Hydrothermal method, thermal treatment	1 M KOH	OER	2	250@20 mA cm^−2^	85	[49]
			HER		147@20 mA cm^−2^	101	
WO_3_·2H_2_O/WS_2_	Anodic treatment	0.5 M H_2_SO_4_	HER	2	152@100 mA cm^−2^	54	[141]
Ni_2_P-WO_3_	Electrodeposition process, phosphating treatment	0.5 M H_2_SO_4_	HER	0.5	107	55.9	[151]
		1 M KOH			105	64.2	
a-WO_x_/WC	Carburization process	0.5 M H_2_SO_4_	HER	5	233@20 mA cm^−2^	69	[52]
		0.1 M PB			292@20 mA cm^−2^	82	
R-Ni_17_W_3_/WO_2_	Hydrothermal method, thermal treatment	1 M KOH	HER	5	48	33	[152]
W_17_O_47_-MoS_2_	Wet chemical, thermal treatment	0.5 M H_2_SO_4_	HER	10	145	41	[57]
WO_2_@C_3_N_4_	Hydrogen thermal process and calcination	0.5 M H_2_SO_4_	HER	5	98	94.4	[36]
MoS_2_@dWO_3_	In situ wet etching	0.5 M H_2_SO_4_	HER	2	191	42	[153]
Fe_2_P-WO_2.92_	Hydrothermal method, thermal treatment	1 M KOH	OER	0.2	215	46.3	[72]
O_v_-WO_x_@NC-Ni	Precursor growth and pyrolysis reaction	1 M KOH	HER	5	67@20 mA cm^−2^	62	[154]
WS_2_/WO_x_/C	Spin-coated and thermal treatment	0.5 M H_2_SO_4_	HER	5	120	36	[155]
Pt@WO_3_/C	Microemulsion method and annealing treatment	1 M PBS	HER	5	149	150	[156]
Ru_2_P/WO_3_/NPC	Hydrothermal method, thermal treatment	1 M KOH	HER	5	15	18	[157]
(Co_5.47_N-CO_2_)@C	Hydrothermal method, thermal treatment	1 M KOH	HER	5	36	68	[158]
			OER		270	60	
			OWS		1.54	-	
WO_3_@F-GS	Plasma-induced assembly method	1 M KOH	OER	5	298	77.6	[133]
CoSe_2_/WSe_2_/WO_3_ NWs	Hydrothermal method, thermal treatment	1 M KOH	HER	5	115	121	[159]
Ru/N-doped Carbon@WO_3_-W_2_C	Co-precipitation method and polyol reduction method	1 M KOH	HER	-	31	40.5	[160]
NiWO_4_-WO_3_-WO_2.9_	Electrospinning method, thermal treatment	0.5 M H_2_SO_4_	HER	2	40	69	[53]
		0.1 M KOH			50	45	
rGO/WS_2_/WO_3_	One-step aerosolization	0.5 M H_2_SO_4_	HER	-	113	37	[161]
WNi_4_@W-WO_2_	Hydrothermal method, thermal treatment	1 M KOH	HER	-	83	83	[162]
Pt_2_W/WO_3_/RGO	Co-deposition	0.5 M H_2_SO_4_	HER	5	394@500 mA cm^−2^	36.8	[163]
2D-Ni(Co,Fe)P/1D-WO_x_	Thermal evaporation– electrodeposition–thermal treatment	1 M KOH	HER	2	49	108	[164]
			OER		270	351	
			OWS		1.51 V	-	
N-doped graphene/WSe_2_-WO_3_	Liquid-phase physical co-exfoliation and heat treatment	0.5 M H_2_SO_4_	HER	5	257	101.9	[165]
WO_2_-Na_x_WO_3_@FeOOH/NF	Hydrothermal method, thermal treatment, FeOOH coating	1 M KOH	OER	1	220@20 mA cm^−2^	42.2	[166]
TA-Fe@Ni-WO_x_	Hydrothermal method, thermal treatment-FeOOH coating	1 M KOH	HER	2	240@20 mA cm^−2^	63.37	[167]
Fe-WO_x_P/rGO	Hydrothermal, CVD method	0.5 M H_2_SO_4_	HER	2	54.6	41.99	[168]
WO_(3−x)_-WC_y_/CDs	Hydrothermal method	0.5 M H_2_SO_4_	HER	5	65	50	[140]
SiO_2_/WO_3_/NiWO_4_	Electrospinning method, carbonization process	0.5 M H_2_SO_4_	HER	2	320	48	[169]
Pt-WO_3_/rGO aerogel	Solvothermal method, electrodeposition	0.5 M H_2_SO_4_	HER	5	42	30	[71]
WS_2_-WC-WO_3_ NHs	Hydrothermal method	0.5 M H_2_SO_4_	HER	5	352	59	[170]
NC/Vo-WON	Hydrothermal and hydrogenation/nitridation-induced strategy	1 M KOH	HER	2	16	33	[171]
WO_x_-PtNi@Pt DNWs	Wet–chemical method	0.1 M KOH	HER	10	24	-	[55]
		0.1 M HClO_4_			5		
		0.5 M PBS			22		

## Data Availability

In this review manuscript, no new data were created.

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
