# Peer review of "Optimization Methods of Tungsten Oxide-Based Nanostructures as Electrocatalysts for Water Splitting"

_nanomaterials, 2023, doi:10.3390/nano13111727_

Round 1

Reviewer 1 Report

 This study has discussed tungsten oxide-based nanostructures owing to their potential in catalysis, especially in the area of water splitting. However, this work has several drawbacks. I have commented them as follows: 

1-      The article should be proofread by a native English writer. Various parts of the paper should be rewritten. The abstract of the paper should be fully rewritten.

2-      Fig. 4a shows charges on atoms? If so, they are not clarified on the caption. Moreover, it is a unicell of WO3 and how many atoms are there in the cell?

3-      I ask the authors the split most of the figures (Fig. 2-8) to several. Otherwise, the caption of reach figure reads very tedious, and figures are too busy and blurry. Otherwise, they can move some of them as supplementary, after splitting.

4-      In Fig. 4l, the numbers along the x-axis (reaction coordinates) are not shown. Why?

5-      The structure-property relationships are not discussed elaborately. It is essential that the readers of this work should understand what are they?

6-      The conclusion section should summarize the best tungsten-based catalysts identified so far and their effectiveness and limitations.

7-      I cannot see any discussion of reaction paths and related results on specific systems. This is necessary for readers to understand how tungsten-based catalysts are effective for water splitting.

8-      Background references are not upto date!

I have provided my comments to the authors. 

Author Response

Thanks for your letter dated May 9th as well as the reviewers’ reports on our manuscript (No: nanomaterials-2399385). We sincerely appreciate the insightful criticism that reviewers provided on our manuscript. The entire manuscript has undergone careful review in response to your comments and recommendations. The revision details are listed below:

Response to Reviewer #1:

General comment: This study has discussed tungsten oxide-based nanostructures owing to their potential in catalysis, especially in the area of water splitting. However, this work has several drawbacks. I have commented them as follows: 

Comment 1: The article should be proofread by a native English writer. Various parts of the paper should be rewritten. The abstract of the paper should be fully rewritten.

Response: Thank you very much for carefully reading our paper and pointing out its problems. We have repolished our manuscript thoroughly, and all the changes were highlighted in the revised manuscript. Please check.

Comment 2: Fig. 4a shows charges on atoms? If so, they are not clarified on the caption. Moreover, it is a unicell of WO3 and how many atoms are there in the cell?

 Response: The electrons in Fig. 4a are created by removing a neutral O atom from WO3, which can be either paired up in a singlet (closed shell) spin state or unpaired in a triplet spin state. The unicell of WO3 has 32 or 64 atoms in the cell. The reduced WO3-x has been modeled by removing one O atom from the 64-atom (x = 0.06) and 32-atom (x = 0.12) supercells, respectively. Please see the highlighted section on page 10 of the revised manuscript for the details.

Comment 3: I ask the authors the split most of the figures (Fig. 2-8) to several. Otherwise, the caption of reach figure reads very tedious, and figures are too busy and blurry. Otherwise, they can move some of them as supplementary, after splitting.

Response: Thanks for the comment here. Based on your suggestion, we have reorganized the figures and corresponding caption. Please see the Figures of the revised manuscript for the details.

Comment 4: In Fig. 4l, the numbers along the x-axis (reaction coordinates) are not shown. Why?

Response: Thank you very much for carefully reading our paper and pointing out its problems. We added transition state to the figure. Please see the Fig. 6i of the revised manuscript for the details.

Comment 5: The structure-property relationships are not discussed elaborately. It is essential that the readers of this work should understand what are they?

 Response: Thanks for the comment here. We have added more discussion on the structure-property relationship of tungsten oxide-based nanomaterials affected by various strategies in each part, and highlighted the description in the revised manuscript.

Comment 6: The conclusion section should summarize the best tungsten-based catalysts identified so far and their effectiveness and limitations.

Response: Based on your suggestion, we have adjusted the conclusion section. Please see page 20 of the revised manuscript for the details.

Comment 7: I cannot see any discussion of reaction paths and related results on specific systems. This is necessary for readers to understand how tungsten-based catalysts are effective for water splitting.

 Response: Based on your suggestion, we have added the corresponding reaction paths and discussion. Please see the Fig. 3 and Page 5 of the revised manuscript for the details.

Comment 8: Background references are not up to date!

 Response: Based on your suggestion, we have reorganized the introduction section and updated some recent literature, such as [7-11] and [59-63]. Please see the highlighted section on page 1 and page 3 of the revised manuscript for the details.

We hope this version is suitable for Nanomaterials. Please feel free to let us know if you have any inquiries.

Thank you again for these nice suggestions.

With best regards,

Prof. Rongming Wang

Reviewer 2 Report

1.       I suggest that the authors divide the Heterostructure construction section into two or three parts based on the morphology of the composites.

 2.       Please provide two or three more mechanism figures and explanations.

 3.       Once these corrections are made, the manuscript can be accepted.

 4.       Please discuss the importance of this review at the end of the introduction.

Author Response

Thanks for your letter dated May 9th as well as the reviewers’ reports on our manuscript (No: nanomaterials-2399385). We sincerely appreciate the insightful criticism that reviewers provided on our manuscript. The entire manuscript has undergone careful review in response to your comments and recommendations. The revision details are listed below:

Response to Reviewer #2:

Comment 1: I suggest that the authors divide the Heterostructure construction section into two or three parts based on the morphology of the composites.

 Response: Thanks for your suggestion. We have divided the Heterostructure construction section into three parts (semiconductor-WOx, WOx-C, metal-WOx) according to the components. Please see page 14 of the revised manuscript for the details.

Comment 2: Please provide two or three more mechanism figures and explanations.

Response: Based on your suggestion, we have provided the corresponding mechanism figures and explanations. Please see the Fig. 3 and Page 5 of the revised manuscript for the details.

Comment 3: Once these corrections are made, the manuscript can be accepted.

Response: Thank you very much.

Comment 4: Please discuss the importance of this review at the end of the introduction.

Response: Based on your suggestion, we discussed the importance of this review in the introduction of the revised manuscript. Please see page 3 of the revised manuscript for the details.

We hope this version is suitable for Nanomaterials. Please feel free to let us know if you have any inquiries.

Thank you again for these nice suggestions.

With best regards,

Prof. Rongming Wang

Round 2

Reviewer 1 Report

Authors have considered my comments and have revised their paper. I am fine with the current version of the ms, so it may be considered for publication.

............